# Prediction of a Multi-Gene Assay (Oncotype DX and Mammaprint) Recurrence Risk Group Using Machine Learning in Estrogen Receptor-Positive, HER2-Negative Breast Cancer—The BRAIN Study

**DOI:** 10.3390/cancers16040774

**Published:** 2024-02-13

**Authors:** Jung-Hwan Ji, Sung Gwe Ahn, Youngbum Yoo, Shin-Young Park, Joo-Heung Kim, Ji-Yeong Jeong, Seho Park, Ilkyun Lee

**Affiliations:** 1Department of Surgery, International St. Mary’s Hospital, Catholic Kwandong University College of Medicine, Incheon 22711, Republic of Korea; shevchencko@ish.ac.kr; 2Department of Surgery, Gangnam Severance Hospital, Yonsei University College of Medicine, Seoul 06273, Republic of Korea; asg2004@yuhs.ac; 3Department of Surgery, Konkuk University Medical Center, Konkuk University School of Medicine, 120-1 Neungdong-ro, Gwangjin-gu, Seoul 05030, Republic of Korea; 0117652771@kuh.ac.kr; 4Department of Surgery, Inha University Hospital, College of Medicine, Incheon 22332, Republic of Korea; uni140@inha.ac.kr; 5Department of Surgery, Yongin Severance Hospital, Yonsei University College of Medicine, Yongin 16995, Republic of Korea; pianac@yuhs.ac; 6Department of AI Research, Neurodigm, Seoul 04790, Republic of Korea; jjy@neurodigm.ai; 7Division of Breast Surgery, Department of Surgery, Yonsei University College of Medicine, Seoul 03722, Republic of Korea

**Keywords:** breast cancer, multi-gene assay, machine learning, prediction model

## Abstract

**Simple Summary:**

Multi-gene assays (MGAs), such as Oncotype DX and Mammaprint, are used to provide predictive and prognostic values in treatment of ER+HER2− breast cancer. However, their accessibility is restricted due to their high cost in some countries. For this reason, many studies have been conducted to develop the tests that can replace the multi-gene assays, but practicality is still insufficient. The aim of our study is to develop a highly accessible machine learning-based model for predicting the result of MGA. Our accurate and affordable machine learning-based predictive model may serve as a cost-effective alternative to the expensive multi-gene assays.

**Abstract:**

This study aimed to develop a machine learning-based prediction model for predicting multi-gene assay (MGA) risk categories. Patients with estrogen receptor-positive (ER+)/HER2− breast cancer who had undergone Oncotype DX (ODX) or MammaPrint (MMP) were used to develop the prediction model. The development cohort consisted of a total of 2565 patients including 2039 patients tested with ODX and 526 patients tested with MMP. The MMP risk prediction model utilized a single XGBoost model, and the ODX risk prediction model utilized combined LightGBM, CatBoost, and XGBoost models through soft voting. Additionally, the ensemble (MMP + ODX) model combining MMP and ODX utilized CatBoost and XGBoost through soft voting. Ten random samples, corresponding to 10% of the modeling dataset, were extracted, and cross-validation was performed to evaluate the accuracy on each validation set. The accuracy of our predictive models was 84.8% for MMP, 87.9% for ODX, and 86.8% for the ensemble model. In the ensemble cohort, the sensitivity, specificity, and precision for predicting the low-risk category were 0.91, 0.66, and 0.92, respectively. The prediction accuracy exceeded 90% in several subgroups, with the highest prediction accuracy of 95.7% in the subgroup that met Ki-67 <20 and HG 1~2 and premenopausal status. Our machine learning-based predictive model has the potential to complement existing MGAs in ER+/HER2− breast cancer.

## 1. Introduction

In estrogen receptor-positive (ER+), HER2-negative breast cancer, several multi-gene assays (MGAs) are used to provide predictive and prognostic value. Among these, Oncotype DX (ODX; Exact Sciences, Madison, WI, USA) (Genomic Health, Redwood City, CA, USA) and Mammaprint (MMP; Agendia, Amsterdam, The Netherlands) have shown level 1 clinical utility in identifying patients with preserved outcomes when treated with adjuvant endocrine therapy and no chemotherapy through the large, prospective phase 3 clinical trials TAILORx [1] and MINDACT [2]. On this evidence, ODX and MMP are included in the main international clinical guidelines of AJCC 8th edition [3,4,5].

The biggest hurdle that makes MGAs less accessible, such as ODX and MMP, is the high cost of testing. Restrictions on the usual use of MGAs are particularly severe in countries that are not covered by insurance or in developing countries [6,7,8]. If the test is omitted, proper treatment may not be performed because the physicians may not make the right decision. To overcome this problem, the necessity for tests that are relatively inexpensive and can replace MGA has been raised.

For this reason, many alternative tools have been developed to predict ODX risk scores. Several studies have reported statistical models predicting ODX risk groups based on pathological variables, such as Magee equations (MEs) [9,10,11], IHC4 score [12], or Adjuvant! Online. In addition, several studies have reported on models that predict MGA risk based on MRI imaging features [13]. Some alternative tests are being used, but they are insufficient to completely replace MGAs.

Recently, with the development of AI (artificial intelligence), several studies have reported the development of models to predict ODX risk based on deep learning [14,15,16]. However, there have not been many studies about AI predictive models. In addition, they have limitations in terms of accuracy, and the amount of data used in each study was low.

The purpose of our study was to develop a more accurate and affordable prediction model using machine learning for high-risk or non-high-risk groups according to the TAILORx [1] and MINDACT criteria [2]. (i.e., recurrent score > 25 for high risk in ODX; MMP index < 0 for high risk in MMP). This study is collaborative research involving multiple institutions, and it utilizes a larger number of cohorts compared to previous studies. While previous studies have focused on predicting ODX among MGAs, this study is the first attempt to also include prediction of MMP and ensemble (MMP + ODX) in addition to prediction of ODX using machine learning.

### Related Work

Several AI-based models have been reported to predict ODX risk using various clinicopathologic features in ER+, HER2-negative breast cancer. Xiaoxian et al. developed a linear regression model to predict the ODX risk category by utilizing deep learning-derived image features and Magee features [17]. They used the Mask R-CNN to derive the image features such as the tumor cell number, tumor-infiltrating lymphocyte (TIL) number, and nuclear grades from whole-slide images (WSIs) of 382 patients. The concordance rates between the actual RS and their model prediction were 56.1% and 68.0% in each validation set.

Brunetti et al. employed a logistic regression model to predict the ODX risk score using the dynamic contrast-enhanced (DCE) MRI-derived radiomics features of 248 patients [18]. Tumor lesions were manually annotated by three independent operators on DCE-MRI images through 3D region of interest (ROI) positioning. The pandas and scikit-learn Python packages were used for data processing, and a logistic regression machine learning classifier was employed for prediction. In the test set, the accuracy of the model was 63.0%.

Devalland et al. developed a prediction tool for the ODX risk categories based on deep learning and using only the morpho-immuno-histological variables [14]. They built a prediction model using Matlab (R2023a) software with 152 cases for training and 168 cases for testing. Three classifiers were used to learn each ODX risk category. The concordance rate between the actual risk group and the predicted risk group ranged from 53 to 56% for each class.

In the study by Kim et al., they present a random forest model using the Azure machine learning platform, which is a cloud service that enables the execution of machine learning processes, for predicting ODX risk categories [15]. Of the total cases, 208 cases were defined as the modeling group and 76 cases were defined as the validation group. The predictors included 14 clinicopathologic features, including histology, the existence of ductal carcinoma in situ, Ki-67, etc. Then, they trained the modeling set using a two-class decision jungle method for the prediction of the high-risk group and a two-class neural network method for the prediction of the low-risk group. The accuracy of validation was 88% in the high-risk group and 79% in the low-risk group.

The random forest model developed by Pawloski et al. showed performance power with specificity and negative predictive value for identifying low-risk patients at 96.3% and 92.9%. But sensitivity and positive predictive value for predicting high-risk patients were lower (48.3% and 65.1%, respectively) [16]. Their predictive model with 500 trees was developed on the training cohort, using age, tumor size, histology, progesterone receptor (PR) expression, lympho-vascular invasion (LVI), and grade as predictors.

Predictive models using pathology slides or MRI images showed low prediction power and have the disadvantage of requiring additional expert intervention when used in clinical practice. Previous AI-based predictive models using various clinicopathologic features presented an accuracy of 53 to 88%. However, there was a limitation in that the patient data used to develop the model was somewhat small.

Our contribution to this work is two-fold. First, through a multicenter study, we gathered a large cohort of 2565 patients to develop our model. Second, we attempted to develop a model that predicts MMP and ensemble (ODX + MMP) in addition to ODX. As far as we know, AI-based models predicting MMP have not yet been developed.

## 2. Methods

### 2.1. Study Population and Data

Following approval from the Institutional Review Board, we retrospectively identified 2565 breast cancer patients with ER+/HER2− breast cancer, who underwent MGAs (ODX or MMP) and were treated at six institutions in South Korea (The BReast Artificial Intelligence Network group: Catholic Kwandong University International St. Mary’s Hospital, Severance Hospital, Gangnam Severance Hospital, Yongin Severance Hospital, Konkuk University Medical Center, and Inha University Hospital) from May 2008 to May 2023. Patients who received neoadjuvant chemotherapy were excluded. Clinicopathologic data—including age at surgery, height, weight, body mass index (BMI), menarche age, menopausal status, E2, FSH, pathologic tumor size, multiplicity, histologic grade (HG), nuclear grade (NG), extensive intraductal component (EIC), LVI, nodal status, hormone receptor expression level by immunohistochemistry (IHC), and Ki-67 and MGA recurrence score (RS) (ODX and MMP)—were obtained from the patients’ medical records. ER positivity was defined as ≥1% of positive staining cells. HER-2 status was determined via immunohistochemistry and/or fluorescence in situ hybridization. RS results were obtained from the original reports provided by Genomic Health and Agendia.

### 2.2. Statistical Analysis

Continuous variables were compared using Student’s *t*-test. Categorical variables were compared using the χ^2^ test using SPSS (version 22.0; SPSS Inc., Chicago, IL, USA). Receiver operating characteristic (ROC) curves and areas under the ROC curve (AUCs) were calculated. All tests were two-sided, and a *p*-value of <0.05 was considered statistically significant.

### 2.3. Data Preprocessing and Feature Selection

We performed a thorough exploratory analysis of the data, addressing outliers introduced during collection. To enhance data quality, we corrected abnormal data points resulting from errors and anomalies. Additionally, we utilized specific techniques like imputation with median, mean, and mode, tailored to each feature’s characteristics, to fill in missing values in the dataset. Additionally, we used MinMaxScaler to scale the data, ensuring that each feature had a consistent impact by transforming their range to fall within 0 to 1. This normalization process facilitates fair comparisons across all features. Next, we carefully selected the predictors by considering factors such as the availability of valid data points, data diversity, and the suitability of each feature for predicting MGA risk. In total, 22 features were selected as model predictors including age at surgery, height, weight, BMI, menarche age, menopausal status, E2, FSH, pathologic tumor size, multiplicity, HG, NG, EIC, LVI, number of extracted sentinel lymph nodes (SLNs), number of metastatic SLNs, perinodal extension, ER, PR, HER2, and Ki-67 and MGA recurrence score. This selection process aimed to ensure a comprehensive and effective set of features for our analysis. Subsequently, MMP and ODX data were separated for individual use, and the modeling and test datasets were divided into 85% and 15% of the total, respectively. As a result, the MMP dataset was divided into 447 patients for the modeling group and 79 patients for the test group, while the ODX dataset was split into 1733 patients for the modeling group and 306 patients for the test group. Combining the MMP and ODX datasets, a total of 2180 patients formed the modeling group, and 385 patients comprised the test group. Specifically, the MMP risk prediction model utilized the MMP dataset, the ODX risk prediction model utilized the ODX dataset, and the ensemble model employed both the MMP and ODX datasets for a comprehensive analysis. The results from the MGAs were categorized to obtain the risk category as a binary outcome (low: RS ≤ 25, high: RS > 25 for ODX; low: index > 0, high: index < 0 for MMP).

### 2.4. AI Modeling and Evaluation

In the AI modeling process (Figure 1), we trained a total of 19 classification models, encompassing various approaches such as a feedforward neural network, to comprehensively assess the performance of the prediction model and efficiently proceed with optimization and ensemble processes. To effectively address the issue of imbalanced data, where the number of instances in each class differs significantly, we employed class-weighted classification techniques during the model training process. This involved assigning higher weights to the minority class to ensure that the model gives adequate consideration to both classes. Additionally, a portion of the modeling dataset, representing 10%, was randomly selected as a validation dataset. Ten random samples were extracted, and cross-validation was performed to evaluate the accuracy of each validation set. The model with the best performance, based on the highest average accuracy across the validation datasets, was then selected. Subsequently, the selected model underwent model optimization using the Hyperopt library. Finally, the MMP risk prediction model utilized a single XGBoost model, and the ODX risk prediction model utilized combined LightGBM.

Additionally, the ensemble model combining MMP and ODX utilized CatBoost and XGBoost through soft voting. To address the class imbalance problem between ‘low’ and ‘high’ risk categories, the F1 score was incorporated into the performance metrics for model evaluation, in addition to accuracy. For the final model training, the entire modeling dataset, which includes the combined validation set, was utilized. During the model validation stage, we set the ‘low risk’ as a ‘positive’ class and evaluated the overall performance on 11 randomly selected modeling–test dataset pairs. Following this assessment, we chose the final modeling–test dataset pair by selecting the one with the median of the performance metrics across all evaluated dataset pairs. This approach ensured that the selected pair represented a balanced and representative performance assessment, contributing to the robustness of our model evaluation process.

## 3. Results

### 3.1. Patient and Tumor Characteristics

The development cohort consisted of a total of 2565 patients, including 2039 patients tested with ODX and 526 patients tested with MMP, respectively. The median patient age was 50 years of age, and the median tumor size was 1.5 cm. Among tumors for which the Nottingham grade was reported (99.6%; *n* = 2556), 704 (27.5%) were low grade, 1617 (63.3%) were intermediate grade, and 235 (9.2%) were high grade. LVI was absent in 2196 cases (86.0%). All tumors (100%) were ER+, and 129 tumors (5.0%) were weakly positive via IHC (Allred score ≤ 5).

In the ensemble cohort, the modeling and test groups consisted of 2180 (85%) patients and 385 (15%) patients, respectively. The modeling and test groups consisted of 447 patients and 79 patients in the MMP cohort and 1733 patients and 306 patients in the ODX cohort. The proportion of low-risk patients was higher in the modeling group than in the test group. However, there were no statistical differences between the modeling group and the test group.

Between the MGA risk groups, several factors showed significant differences. In the MMP cohort, the low-risk group exhibited smaller tumor size, lower HG/NG, and lower Ki-67 levels compared to the high-risk group (Table 1). In the ODX cohort, the low-risk group had a younger age at diagnosis, earlier menarche age, and a higher percentage of patients with premenopausal status than the high-risk group. Additionally, there were significant differences between the risk groups in terms of preoperative E2/FSH levels, tumor size, ER/PR/HER2 expression, HG/NG, and Ki-67 (Table 2). The ensemble cohort showed similar results to the ODX cohort.

In the ODX cohort (*n* = 2039), the median observed RS was 16. Only 14.6% of patients (*n* = 297) were classified as high risk (RS > 25). However, in the MMP cohort (*n* = 526), 36.3% of the patients were classified as high risk (*n* = 191) (Figure 2). Thus, an imbalance in risk distribution between the MGAs was identified.

### 3.2. Machine Learning Model Prediction of MGA Risk Category

Our study identified that the accuracy of our predictive model was 86.8% for the ensemble model, 84.8% for the MMP model, and 87.9% for the ODX model (Table 3). In the ensemble cohort, the sensitivity, specificity, and precision for predicting the low-risk category were 0.91, 0.66, and 0.92, respectively (Figure 3 and Table 4). The sensitivity, specificity, and precision of the MMP prediction model were 0.86, 0.83, and 0.92. Those of the ODX prediction model were 0.93, 0.58, and 0.92, respectively (Table 4). Additionally, the AUC of the ROC curve was 0.86 in the ensemble cohort (Figure 4).

### 3.3. Cross-Tab Analysis and Subgroup Analysis

We analyzed the data of patients who had undergone MMP using a predictive model based on ODX. Likewise, the opposite analysis was also implemented. The results of the cross-tab analysis of predicting ODX data using the MMP predictive model and MMP data using the ODX predictive model are presented in Table 5.

We conducted a subgroup analysis according to the various clinical situations. As a result, it was confirmed that the prediction accuracy exceeded 90% in several subgroups. The accuracy of the MMP prediction model for postmenopausal patients was 93%. Furthermore, the accuracy of the ensemble prediction model was the highest at 95.7% in the case of PR positivity, Ki-67 levels < 20, and premenopausal status (Appendix A).

### 3.4. Analysis of Model-Underpredicted (Discordant) Cases

We also identified the characteristics of model-underpredicted (discordant) cases. Compared with the prediction success group, the prediction fail group exhibited older age, a higher FSH level, larger tumor size, higher grades, more LN metastasis, a higher PR negative rate, more HER2 expression, a higher Ki-67 level, and higher MGA risk (Appendix A).

Interestingly, most discordant cases were distributed in border values between low risk and high risk. In the ODX cohort, 62.2% (23/37) of discordant cases had ODX risk score values within the range of 20 to 32. Furthermore, in the MMP cohort, 83.3% (10/12) of discordant cases had an MMP index within the range of −0.2 to 0.2 (Figure 5).

## 4. Discussion

This study included ER+HER2− breast cancer patients who had undergone MMP or ODX. While many studies have been conducted to develop models that can predict ODX outcomes, none have ventured into developing models for predicting MMP. In this respect, our study stands as a novel attempt and has the largest number of cohorts compared to previous studies. Statistically significant factors between the low- and high-risk groups in the MMP and ODX cohorts are the same as those previously known (Table 1 and Table 2).

Our cohort exhibits an imbalance between the low-risk group and the high-risk group. In the MMP cohort (*n* = 526), 36.3% of patients were classified as high risk, while in the ODX cohort (*n* = 2039), 14.6% of patients were categorized as high risk. We analyzed these cohort data by adding an F1 score indicator in addition to accuracy to overcome the imbalance problem. The imbalance in data means the presence of selection bias because MGA is not applied to all patients with HR+HER2− breast cancer in the real clinical field. Based on these data, nevertheless, the accuracy of our predictive model was 86.8%, 84.8%, and 87.9% in the ensemble cohort, MMP cohort, and ODX cohort, respectively (Table 3). In previous studies predicting only ODX, the predictive model with logistic regression developed by Orucevic et al. showed an overall accuracy of 86.8% [19]. They developed a predictive model using a sample of approximately 65,754 patients from the National Cancer Database, which was predictive of low- and high-risk categories and was highly sensitive (99%) for predicting RS ≤ 25 in all ages [19]. Some studies have reported machine learning-based predictive models to predict ODX risk categories using clinicopathologic data. Kim et al. developed a random forest model using the Azure machine learning platform for predicting ODX risk categories. The accuracy of validation was 88% in the high-risk group and 79% in the low-risk group. The AUC of the ROC curve was 0.917 in the high-risk group and 0.744 in the low-risk group in the test cohort [15]. The predictive model developed by Pawloski et al. showed performance power with specificity and negative predictive value for identifying low-risk patients at 96.3% and 92.9%. But the sensitivity and positive predictive value for predicting high-risk patients were lower (48.3% and 65.1%, respectively) [16]. Previous studies have divided data into low- and high-risk groups and developed models that predict each risk group. However, our study is differentiated by developing a model for predicting MGA risk in the entire data containing both low- and high-risk groups. Nevertheless, our model did not have a lower prediction rate and demonstrated comparable performance power.

The recent development of commercially available MGAs has proven to be prognostic, with data supporting their ability to predict chemotherapy benefits. Consensus guidelines support the routine use of these tests to guide adjuvant therapeutic decisions for eligible patients worldwide [20,21]. However, it is common knowledge that results can vary significantly even between different MGAs [22]. Prat A. et al. reported that various genomic signatures including PAM50-ROR, MMP, and ODX often produce discordant risk results even in the same patient cohort [23]. Intriguingly, high discordance rates have been reported even between ODX and MMP, both of which are included in the major international clinical guidelines. Over 30% of MMP high-risk cases were reclassified as low risk by ODX [24]. In this study, we tested the MMP cohort by using a predictive model trained based on ODX data and tested the ODX cohort using a predictive model trained based on MMP data. The cross-tab analysis revealed that the accuracy of the MMP model for the ODX cohort and the ODX model for the MMP cohort was 77.0% and 71.5%, respectively (Table 5). These prediction failure rates are in line with previously reported discordance rates between different MGAs. This means that the previously known discordant rate between MGAs was also confirmed in our cohort by the predictive model.

The reason why these CDSSs (clinical decision support systems) should be continuously developed regardless of existing statistical methods or AI (artificial intelligence) is because of the cost problem, which is the biggest hurdle of MGAs, not only to supplement MGA [25]. Although the cost of ODX and MMP is currently USD 3460 and USD 4250, respectively, a number of retrospective studies have confirmed the cost-effectiveness of these tests in treatment decision-making for breast cancer [26,27,28,29]. However, it remains a limitation that the costs are still high, thus posing challenges in terms of accessibility by nations and individuals [6,7,8]. Reliable identification of patients with low-risk tumors even when MGAs are not readily available is crucial to minimize unnecessary cytotoxic exposure and to reduce the treatment costs [30]. In this regard, Pawloski et al. endeavored to enhance the clinical utility of the developed model by studying cohorts limited to those over the age of 50 [16]. In our ensemble cohort with MGAs implemented, we classified several subgroups and tested the prediction model. We identified that the prediction accuracy exceeded 90% in several subgroups, with the highest prediction accuracy of 95.7% in the subgroup that met Ki-67 < 20 and HG 1~2 and premenopausal status. However, as the test dataset was divided into subgroups, the problem of decreasing the amount of data for testing occurred. If the test dataset itself is insufficient, the results may be unreliable even if the prediction rate is high. The lack of sufficient test datasets for the “High Risk” group in subgroup analysis is a limitation. Therefore, further follow-up research is required to develop models with substantial clinical potential in patients under specific conditions.

Comparison analysis between the prediction success and fail groups identified statistically significant differences in age, menopausal status, FSH, tumor size, HG, NG, LN metastasis, PR, HER2, and Ki-67 (Appendix A). By considering these factors, it may be possible to develop a more improved prediction model and select the appropriate target patient group. In the prediction-failed cases, there were many patients mainly corresponding to boundary values (Figure 5). This phenomenon was also confirmed in a study using a machine learning model by Pawloski et al. [16]. Their model predicted 76 cases as low risk (RS ≤ 25) among 147 women with high-risk tumors (RS > 25). Of these cases, the median observed RS was 29 (IQR 27–31). Over 71% of these tumors had RS results between 26 and 30 (*n* = 54).

One of the strengths of our study is the large sample size drawn from a multicenter database for modeling a predictive model. Moreover, we developed the predictive model using various non-routine parameters in addition to clinicopathologic variables related to breast cancer prognosis. As mentioned earlier, our model was developed to predict not only ODX but also MMP risk category.

However, our study has several limitations. First of all, because our study was performed by multiple centers, there may have been measurement deviations in the data collection. However, this situation can serve as both a disadvantage and an advantage. Our models were developed under these circumstances, which may ultimately reduce these limitations when extending coverage to various clinical institutions. Second, the amount of data used in the subgroup analysis was small. Third, our ensemble predictive model in this study might be affected by data bias by using two MGAs. Lastly, our study is retrospective in nature.

## 5. Conclusions

We have developed a machine learning-based predictive model that has the potential to complement existing MGAs in ER+HER2− breast cancer. We identified that the accuracy of the ensemble predictive model was 86.8% overall, especially 95.7% in the case of PR positivity, Ki-67 levels < 20, and premenopausal status. Although the accuracy of our model is still somewhat insufficient to apply to all ER+HER2− breast cancer patients, it is expected to be feasible in a well-selected group with variables such as PR positivity, Ki-67 levels < 20, and premenopausal status. Currently, we are planning a prospective study based on our model with external validation. As more patient data are accumulated and survival data are added in the future, it is expected that a more accurate and widely available predictive model could be developed.

## Figures and Tables

**Figure 1 cancers-16-00774-f001:**
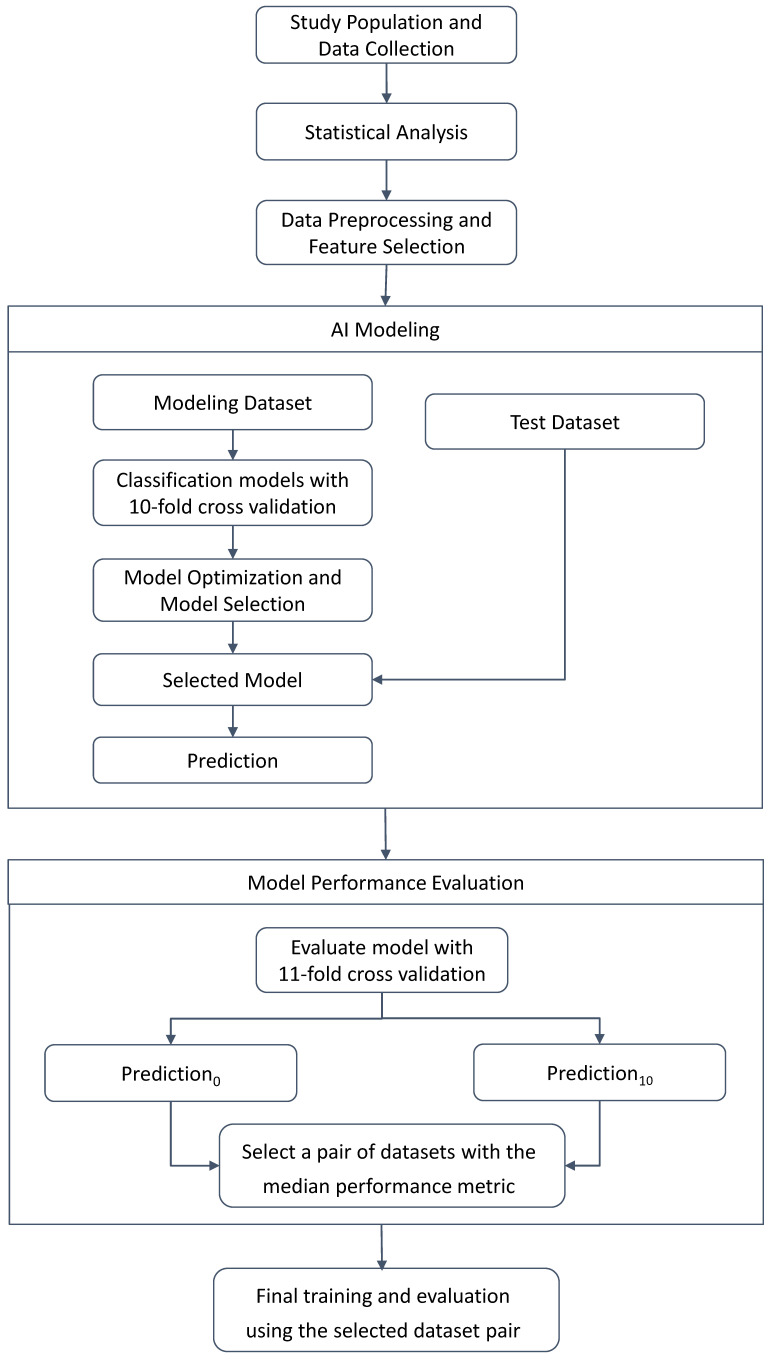
The workflow of modeling using machine learning.

**Figure 2 cancers-16-00774-f002:**
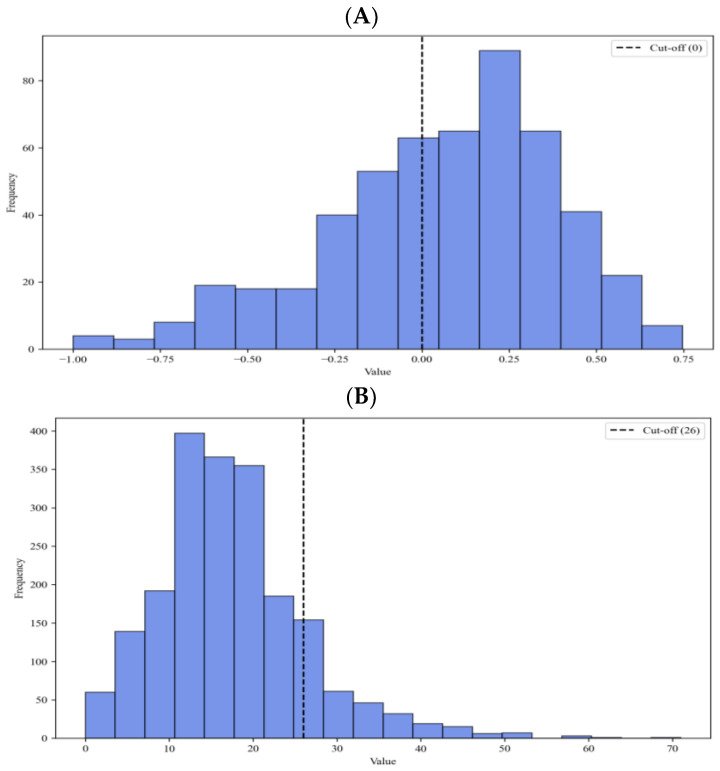
Distribution of MGA scores. (**A**) In the MMP cohort (*n* = 526), 36.3% of patients (*n* = 191) were classified as high risk (MMP index < 0). (**B**) In the ODX cohort (*n* = 2039), 14.3% of patients (*n* = 297) were classified as high risk (ODX risk score > 25). MGA, multi-gene assay; MMP, Mammaprint; ODX, Oncotype DX.

**Figure 3 cancers-16-00774-f003:**
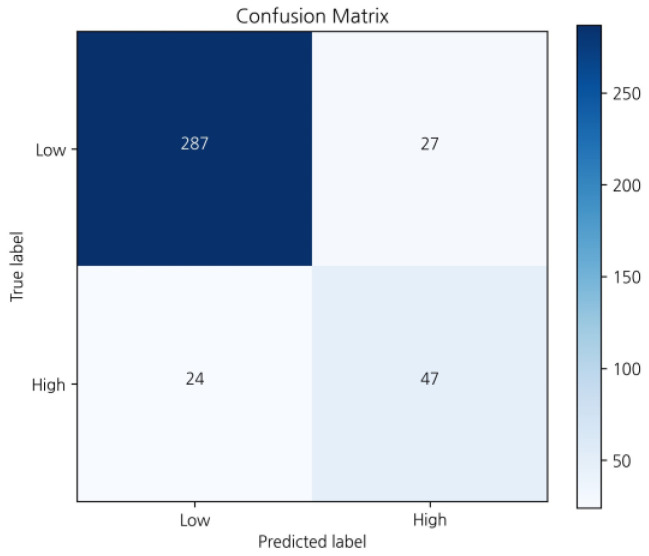
Confusion matrix of prediction performance in the ensemble cohort.

**Figure 4 cancers-16-00774-f004:**
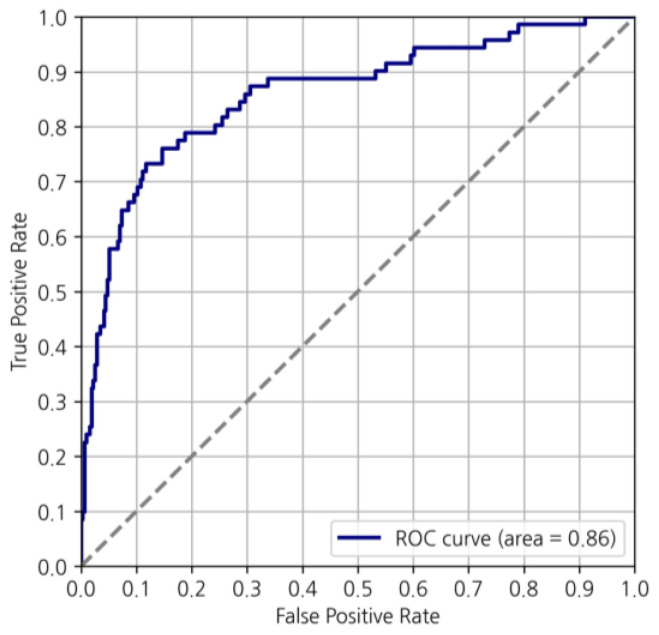
The mean AUC of the ROC curve for the high-risk group and the low-risk group in the ensemble cohort. AUC, area under the curve; ROC, receiver operating characteristic.

**Figure 5 cancers-16-00774-f005:**
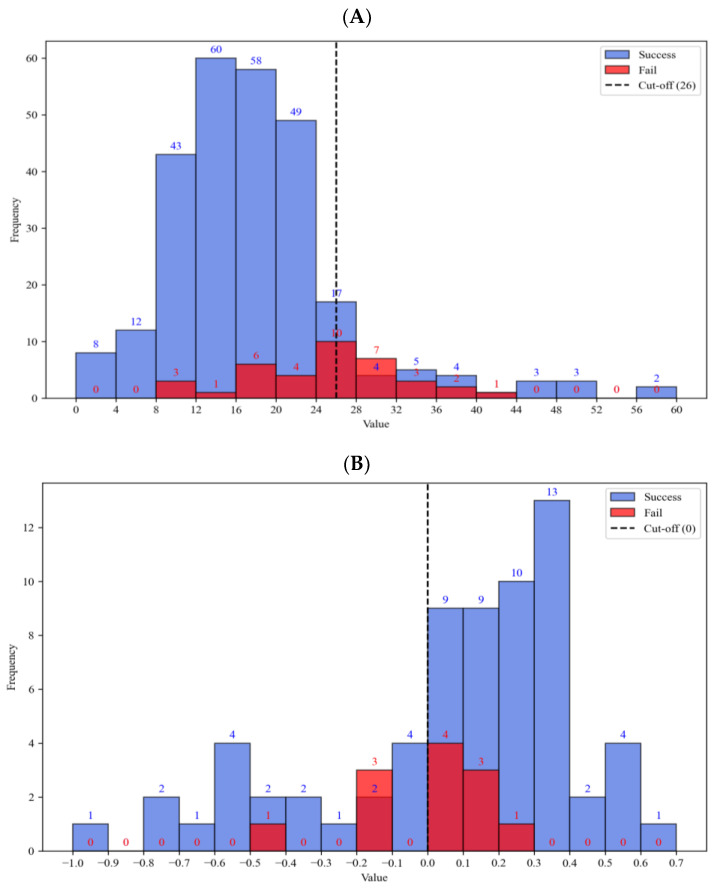
Distribution of MGA values according to the prediction results. (**A**) In the ODX cohort, 62.2% (23/37) of discordant cases had boundary ODX risk scores. (**B**) In the MMP cohort, 83.3% (10/12) of discordant cases had a boundary MMP index. MGA, multi-gene assay; ODX, Oncotype DX; MMP, Mammaprint.

**Table 1 cancers-16-00774-t001:** Baseline clinicopathologic characteristics between the low- and high-risk groups in the MMP cohort.

Variables	MMP (*n* = 526)		
	Low (*n* = 335)	High (*n* = 191)	*p*-Value
Age (mean ± SD)	53.27 ± 9.07	52.62 ± 10.44	0.473
Height (mean ± SD)	159.23 ± 5.41	158.81 ± 5.06	0.380
Weight (mean ± SD)	60.54 ± 10.09	60.62 ± 8.09	0.920
BMI (mean ± SD)	23.87 ± 3.78	24.05 ± 3.10	0.590
Menarche age (mean ± SD)	14.41 ± 1.76	14.42 ± 1.57	0.933
Menopausal status			0.656
Premenopausal	176 (52.5%)	96 (50.3%)	
Postmenopausal	158 (47.2%)	95 (49.7%)	
Unknown	1 (0.3%)	0 (0.0%)	
Preoperative E2 (mean ± SD)	73.15 ± 98.06	77.01 ± 98.34	0.676
Preoperative FSH (mean ± SD)	40.93 ± 36.29	38.24 ± 34.68	0.422
Tumor size (mean ± SD)	1.62 ± 0.65	1.97 ± 0.82	<0.001
Multiple lesions			0.213
No	251 (74.9%)	153 (80.1%)	
Yes	84 (25.1%)	38 (19.9%)	
HG			<0.001
1	123 (36.7%)	30 (15.7%)	
2	197 (58.8%)	101 (52.9%)	
3	12 (3.6%)	59 (30.9%)	
Unknown	3 (0.9%)	1 (0.5%)	
NG			<0.001
1	32 (9.6%)	3 (1.6%)	
2	274 (81.7%)	134 (70.2%)	
3	28 (8.4%)	53 (27.7%)	
Unknown	1 (0.3%)	1 (0.5%)	
EIC			0.131
No	218 (65.1%)	138 (72.2%)	
Yes	108 (32.2%)	46 (24.1%)	
Unknown	9 (2.7%)	7 (3.7%)	
LVI			0.192
No	285 (85.1%)	153 (80.1%)	
Yes	49 (14.6%)	37 (19.4%)	
Unknown	1 (0.3%)	1 (0.5%)	
SLN (mean ± SD)	2.55±1.60	2.80±2.50	0.172
Lymph node metastasis			0.352
No	182 (54.3%)	93 (48.7%)	
Yes	148 (44.2%)	93 (48.7%)	
Unknown	5 (1.5%)	5 (2.6%)	
Perinodal extension			0.545
No	288 (86.0%)	169 (88.5%)	
Yes	46 (13.7%)	22 (11.5%)	
Unknown	1 (0.3%)	0 (0.0%)	
Estrogen receptor			0.271
Low (0~5)	11 (3.3%)	10 (5.2%)	
High (6~8)	324 (96.7%)	181 (94.8%)	
Progesterone receptor			0.112
Negative	39 (11.6%)	34 (17.8%)	
Positive	296 (88.4%)	157 (82.2%)	
HER2 receptor			0.062
0	77 (23.0%)	50 (26.2%)	
1+	156 (46.6%)	69 (36.1%)	
2+	102 (30.4%)	72 (37.7%)	
Ki-67 (mean ± SD)	13.17 ± 11.73	27.96 ± 18.89	<0.001

BMI, body mass index; E2, estradiol; FSH, follicle-stimulating hormone; HG, histologic grade; NG, nuclear grade; EIC, extensive intraductal component; LVI, lympho-vascular invasion; SLN, sentinel lymph node; HER2, human epidermal growth factor receptor 2.

**Table 2 cancers-16-00774-t002:** Baseline clinicopathologic characteristics between the low- and high-risk groups in the ODX cohort.

Variables	ODX (*n* = 2039)		
	Low (*n* = 1742)	High (*n* = 297)	*p*-Value
Age (mean ± SD)	50.21 ± 9.52	52.76 ± 9.81	<0.001
Height (mean ± SD)	159.31 ± 5.34	158.88 ± 5.35	0.201
Weight (mean ± SD)	58.77 ± 8.31	58.25 ± 8.22	0.315
BMI (mean ± SD)	23.15 ± 3.27	23.01 ± 3.21	0.516
Menarche age (mean ± SD)	14.18 ± 1.56	14.60 ± 1.70	<0.001
Menopausal status			<0.001
Premenopausal	1061 (60.9%)	122 (41.1%)	
Postmenopausal	671 (38.5%)	174 (58.6%)	
Unknown	10 (0.6%)	1 (0.3%)	
Preoperative E2 (mean ± SD)	107.12 ± 149.39	72.06 ± 103.40	<0.001
Preoperative FSH (mean ± SD)	30.68 ± 33.55	44.22 ± 37.24	<0.001
Tumor size (mean ± SD)	1.62 ± 0.72	1.77 ± 0.67	0.001
Multiple lesions			0.191
No	1320 (75.8%)	236 (79.5%)	
Yes	422 (24.2%)	61 (20.5%)	
HG			<0.001
1	522 (30.0%)	29 (9.8%)	
2	1124 (64.5%)	195 (65.7%)	
3	92 (5.3%)	72 (24.2%)	
Unknown	4 (0.2%)	1 (0.3%)	
NG			<0.001
1	134 (7.6%)	7 (2.4%)	
2	1419 (81.5%)	187 (63.0%)	
3	184 (10.6%)	102 (34.3%)	
Unknown	5 (0.3%)	1 (0.3%)	
EIC			0.293
No	934 (53.6%)	175 (58.9%)	
Yes	568 (32.6%)	91 (30.7%)	
Unknown	240 (13.8%)	31 (10.4%)	
LVI			0.559
No	1498 (86.0%)	260 (87.5%)	
Yes	236 (13.5%)	36 (12.1%)	
Unknown	8 (0.5%)	1 (0.3%)	
SLN (mean ± SD)	2.81±2.05	2.64±1.67	0.161
Lymph node metastasis			0.093
No	1531 (87.9%)	273 (91.9%)	
Yes	189 (10.8%)	23 (7.8%)	
Unknown	22 (1.3%)	1 (0.3%)	
Perinodal extension			0.430
No	1571 (90.2%)	273 (91.9%)	
Yes	39 (2.2%)	4 (1.3%)	
Unknown	132 (7.6%)	20 (6.8%)	
Estrogen receptor			0.004
Low (0~5)	82 (4.7%)	26 (8.8%)	
High (6~8)	1660 (95.3%)	271 (91.2%)	
Progesterone receptor			<0.001
Negative	193 (11.1%)	109 (36.7%)	
Positive	1549 (88.9%)	188 (63.3%)	
HER2 receptor			<0.001
0	506 (29.0%)	78 (26.3%)	
1+	805 (46.2%)	113 (38.0%)	
2+	431 (24.8%)	106 (35.7%)	
Ki-67 (mean ± SD)	13.91 ± 11.64	24.93 ± 16.68	<0.001

BMI, body mass index; E2, estradiol; FSH, follicle-stimulating hormone; HG, histologic grade; NG, nuclear grade; EIC, extensive intraductal component; LVI, lympho-vascular invasion; SLN, sentinel lymph node; HER2, human epidermal growth factor receptor 2.

**Table 3 cancers-16-00774-t003:** The predictive result according to MGAs.

Cohort	Modeling/Test	Accuracy	F1 Score
MMP	447/79	84.8%	0.8889
ODX	1733/306	87.9%	0.9287
Ensemble (MMP + ODX)	2180/385	86.8%	0.9184

MMP, Mammaprint; ODX, Oncotype DX.

**Table 4 cancers-16-00774-t004:** The predictive result of the test cohort.

MGA Cohort	Sensitivity	Specificity	Precision	F1 Score	Accuracy
MMP	0.8571	0.8261	0.9231	0.8889	0.848
ODX	0.9341	0.5833	0.9234	0.9287	0.879
Ensemble	0.9140	0.6620	0.9228	0.9184	0.868

MGA, multi-gene assay; MMP, Mammaprint; ODX, Oncotype DX.

**Table 5 cancers-16-00774-t005:** Cross-tab analysis.

Modeling Dataset	Test Dataset	Modeling/Test	Accuracy	F1 Score
MMP data	ODX data	447/2039	77.0%	0.8584
ODX data	MMP data	1733/526	71.5%	0.8077

MMP, Mammaprint; ODX, Oncotype DX.

## Data Availability

The datasets used and/or analyzed during the current study are available from the corresponding author upon reasonable request.

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
