# Peer review of "Prediction of a Multi-Gene Assay (Oncotype DX and Mammaprint) Recurrence Risk Group Using Machine Learning in Estrogen Receptor-Positive, HER2-Negative Breast Cancer—The BRAIN Study"

_cancers, 2024, doi:10.3390/cancers16040774_

Round 1
Reviewer 1 Report
Comments and Suggestions for Authors
The manuscript is well-organized and scientifically sound. Everything is a perfect discussion which is too long and it is advice to add a related work section.
Author Response
Comment: We compare your manuscript with published materials and there are some overlaps should be rewritten.
Response: Some parts that overlap with published material have been deleted or rewritten.
(Reviewer 1)
Comment: The manuscript is well-organized and scientifically sound. Everything is a perfect discussion which is too long and it is advice to add a related work section
Response: Thank you for your comment. As you suggested, we have simplified the content of the discussion.
Reviewer 2 Report
Comments and Suggestions for Authors
This research focused on constructing machine learning-based models to anticipate risk categories in multi-gene assays for ER+/HER2− breast cancer patients who underwent either Oncotype DX or MammaPrint tests. Analyzing a group of 2,565 patients, the authors show that the models achieved an overall accuracy rate of 87.3%. Specifically, the MammaPrint model exhibited an 84.8% accuracy, while the combined Oncotype DX model, utilizing LightGBM, CatBoost, and XGBoost, achieved 87.9%. Authors suggests that these machine learning-driven models hold promise as supplementary tools for assessing multi-gene assays in ER+/HER2− breast cancer cases.
Overall, the study has used significant number of patient data points to build the machine learning model and presented their utility however, there are a few minor concerns which authors could address to further clarify the utility of their model, w.r.t. the patient outcome and treatment advantage.
Comments:
1. Did the authors seek advice from a physician regarding the optimal sensitivity and specificity thresholds for this predictive model to positively influence patient outcomes?
2. Given the discussion by the authors highlighting other models with higher accuracy, what advantages does the model in this manuscript offer compared to existing ones, specifically in benefiting patients? How should a physician navigate the selection of a model among several available methods?
3. The authors have addressed the economic burden and access issues related to multi-gene assays (MGA). How does the use of the predictive model in this manuscript offer a solution to these challenges? Can the predictive model reach similar conclusions without the need for an MGA, relying on other factors? If an MGA remains necessary, how would the predictive model still provide economic benefits?
Author Response
Comment: Did the authors seek advice from a physician regarding the optimal sensitivity and specificity thresholds for this predictive model to positively influence patient outcomes?
Response: In general, there is no specific threshold of accuracy that is clearly recommended for the new test. Many tools are being studied to replace the ODX, and in the case of Geneswell (BCT), one of them, the concordance rate with ODX was reported as 71.9%*, which is actually used clinically in South korea. Our model showed an accuracy of 86.8%, which considered to be a clinically acceptable range.
* Kwon, M.J., et al., Comparison of GenesWell BCT Score With Oncotype DX Recurrence Score for Risk Classification in Asian Women With Hormone Receptor-Positive, HER2-Negative Early Breast Cancer. Front Oncol, 2019. 9: p. 667.
Comment: Given the discussion by the authors highlighting other models with higher accuracy, what advantages does the model in this manuscript offer compared to existing ones, specifically in benefiting patients? How should a physician navigate the selection of a model among several available methods?
Response: What kind of MGA to use in actual clinical practice can vary depending on the clinician's propensity or medical environment. Our ensemble model was developed by integrating patients who underwent ODX and MMP, so it is expected to be available in a more comprehensive patient group. If studies on these AI-based prediction models continues in the future, physicians will prefer models with more evidence.
Comment: The authors have addressed the economic burden and access issues related to multi-gene assays (MGA). How does the use of the predictive model in this manuscript offer a solution to these challenges? Can the predictive model reach similar conclusions without the need for an MGA, relying on other factors? If an MGA remains necessary, how would the predictive model still provide economic benefits?
Response: The accuracy of our model is 86.8%, which does not show a perfect match rate with the existing MGA. However, certain subgroups showed accuracy of nearly 95% (Ki-67 <20 and HG 1~2 and premenopausal status). Based on this, in situations where it is difficult to implement MGA, it is expected that our model can be used as a replacement tool, especially in specific subgroup.
Reviewer 3 Report
Comments and Suggestions for Authors
In the abstract:
There are acronyms not previously specified.
The type of data on which the forecasting algorithms were trained is not indicated.
It is not clear what overall accuracy refers to. To specify.
In general, the validation technique used is not indicated and therefore what the indicated performances refer to.
The state of the art regarding the development of prediction models using clinical characteristics is not complete. There are ML models that predict the risk of disease recurrence starting from the clinical characteristics knowing the follow-up of the patients. For example:
Fanizzi, A., et. to the. (2023). Machine learning survival models trained on clinical data to identify high risk patients with hormone responsive HER2 negative breast cancer. Scientific Reports, 13(1), 8575.
Wu, X. et al. Personalized prognostic prediction models for breast cancer recurrence and survival incorporating multidimensional data. JNCI J. Nat. Cancer Institute 109(7), 314 (2017).
Laas, E. et al. Are we able to predict survival in ER-positive HER2-negative breast cancer? A comparison of web-based models. Br J Cancer 112(5), 912–917 (2015).
Massafra, R. et al. A clinical decision support system for predicting invasive breast cancer recurrence: Preliminary results. Front. Oncol. 11, 1–13 (2021).
Even in the introduction the type of data used to train the models is not indicated.
In the introduction the authors declare that “..this study 74 is the first attempt to simultaneously predict both ODX and MMP using machine learning”. However, in the abstract they state that “
Patients with ER+/HER2− breast cancer, who underwent 31 Oncotype DX or MammaPrint were used to develop the prediction model. Please clarify further. Does the model predict both outcomes even though the samples were independent? If this were the case, the hypothesis of concordance between the results of different genomic tests carried out on the same sample of patients should be highlighted in the introduction with appropriate bibliographical references.
In general terms, it is suggested to describe the methods used in much more detail, from pre-processing to validation.
‘We conducted exploratory analysis of the data, processed outliers, and supplemented missing values.’ Describe in more detail the techniques used to process outliers and replace missing data.
Clarify better how the 22 features were selected.
In section 2.4 clarify the statement 'In the AI modeling process, a total of 19 different classification models were utilised, 112 including deep learning models'
Specify which class weighted classification techniques were employed during the model training process.
Clarify the statement 'Finally, to validate the model's performance, we 124 used the 11 randomly selected test datasets, with the median of the performance metrics 125 serving as the criteria.'
I believe that the development of a model trained on the entire sample of heterogeneous patients in terms of risk assessment (outcome of interest assessed with different genomic tests) is risky if significant concordance between the genomic tests considered on the basis of data is not verified in the literature of literature. This is my main concern. For the same reason, I do not consider the analysis referred to in the tab to be valid. 5. I would suggest keeping the analyzes separate.
The results report the following: 'In the ensemble cohort (MMP + ODX), the 165 sensitivity, specificity, and precision for predicting the low-risk category were 0.92, 0.66, 166 and 0.92, respectively. And those for predicting the high-risk category was 0.66, 0.92, and 167 0.65, respectively (Figure 2 & Table 4).’ I understand that the models are binary classificatory for predicting the low or high risk category. So why specify sensitivity, specificity and precision for the low-risk class and then for the high-risk class? Clearly define what the positive class (sensitivity and precision) and the negative class (specificity) are. Please explain in the materials section which is the positive reference class and which is the negative one.
Comments on the Quality of English LanguageMinor editing of English language required
Author Response
Comment: (In the abstract) There are acronyms not previously specified.
Response: We added notation for abbreviations.
Comment: (In the abstract) The type of data on which the forecasting algorithms were trained is not indicated.
Response: Thank you for this comment. We have only mentioned the obtained clinicopathologic feature in the method section. Following your suggestion, we stated the specific predictors used in modeling in the method section (lines #107-112). Is it necessary to add this to the abstract as well? If so, let us know and we will add it to the abstract.
Comment: (In the abstract) It is not clear what overall accuracy refers to. To specify.
Response: Thank you for your comment. In fact, we developed a model for predicting ODX and MMP, respectively. And we showed the mean values of both models as overall values. However, as you comment, we’ve considered it again. As a result, we have identified errors for that and integrated ODX and MMP patient data to create a new ensemble model that predicts MGA outcomes. Accordingly, we modified the text as a result of using the ensemble model (lines #39-41).
Comment: (In the abstract) In general, the validation technique used is not indicated and therefore what the indicated performances refer to.
Response: Thank you for your precise comment. We supplemented explanation about the cross-validation technique in Method 2.4. (lines #132-136) and Abstract section (lines #38-39). Additionally, clarifications have been made regarding the process of extracting the validation dataset and selecting the model in Method 2.4. section (line #144-149).
Comment: The state of the art regarding the development of prediction models using clinical characteristics is not complete. There are ML models that predict the risk of disease recurrence starting from the clinical characteristics knowing the follow-up of the patients. For example:
Fanizzi, A., et. to the. (2023). Machine learning survival models trained on clinical data to identify high risk patients with hormone responsive HER2 negative breast cancer. Scientific Reports, 13(1), 8575.
Wu, X. et al. Personalized prognostic prediction models for breast cancer recurrence and survival incorporating multidimensional data. JNCI J. Nat. Cancer Institute 109(7), 314 (2017).
Laas, E. et al. Are we able to predict survival in ER-positive HER2-negative breast cancer? A comparison of web-based models. Br J Cancer 112(5), 912–917 (2015).
Massafra, R. et al. A clinical decision support system for predicting invasive breast cancer recurrence: Preliminary results. Front. Oncol. 11, 1–13 (2021).
Response: We totally agree with your comment. Unfortunately, however, the aim of this study was to develop a model predicting MGA’s risk groups. I think it will be of great clinical help if the patient's follow-up data is accumulated and an AI model predicting recurrence is developed in the future. The next step of our study is to conduct a prospective study in which long-term outcome (survival, recurrence, etc.) is followed and compared with MGA.
Comment: Even in the introduction the type of data used to train the models is not indicated.
Response: We have only mentioned the obtained clinicopathologic feature in the method section. Following your suggestion, we stated the specific predictors used in modeling (lines #107-112).
Comment: In the introduction the authors declare that “this study is the first attempt to simultaneously predict both ODX and MMP using machine learning”. However, in the abstract they state that “Patients with ER+/HER2− breast cancer, who underwent Oncotype DX or MammaPrint were used to develop the prediction model.” Please clarify further.
Does the model predict both outcomes even though the samples were independent? If this were the case, the hypothesis of concordance between the results of different genomic tests carried out on the same sample of patients should be highlighted in the introduction with appropriate bibliographical references.
Response: We initially developed a model for predicting ODX risk in patients who underwent ODX, and MMP risk in patients who underwent MMP, respectively. However, in accordance with your previous advice, we further developed a new ensemble model that predicts MGA outcomes by integrating data from patients who underwent ODX or MMP. As mentioned in the discussion, it is also known that there is a discrepancy in the results between ODX and MMP as well. But, it is expected that our model will rather help close this gap.
Comment: In general terms, it is suggested to describe the methods used in much more detail, from pre-processing to validation.
Response: In response to the feedback, the paper has been revised to provide more clarity on the process and rationale of data scaling in the 2.3. section. Additionally, the 2.4. section has been updated to enhance clarity regarding the validation process.
Comment: ‘We conducted exploratory analysis of the data, processed outliers, and supplemented missing values.’ Describe in more detail the techniques used to process outliers and replace missing data.
Response: In response to the feedback, the 2.3. section has been revised to provide more detailed information on the techniques employed for processing outliers and imputing missing values.
Comment: Clarify better how the 22 features were selected.
Response: In response to the feedback, the 2.3. section has been revised to provide clearer criteria for the selection of variables. Among the available patient data, factors that have previously been found to be related to the prognosis of breast cancer were selected.
Comment: In section 2.4 clarify the statement 'In the AI modeling process, a total of 19 different classification models were utilised, including deep learning models'
Response: In response to the feedback, the mentioned sentence has been revised to improve clarity about an explanation for the utilization of multiple models.
Comment: Specify which class weighted classification techniques were employed during the model training process.
Response: In response to the feedback, in the 2.4. section, additional details have been provided regarding the class-weighted classification techniques, including an explanation of their application and the rationale behind their use.
Comment: Clarify the statement 'Finally, to validate the model's performance, we used the 11 randomly selected test datasets, with the median of the performance metrics serving as the criteria.'
Response: In response to the feedback, the mentioned sentence has been elaborated to provide more detailed information about the process.
Comment: I believe that the development of a model trained on the entire sample of heterogeneous patients in terms of risk assessment (outcome of interest assessed with different genomic tests) is risky if significant concordance between the genomic tests considered on the basis of data is not verified in the literature of literature. This is my main concern. For the same reason, I do not consider the analysis referred to in the tab to be valid. I would suggest keeping the analyzes separate.
Response: It is well known that in actual clinical practice, various MGAs are performed according to the clinician's choice, and even between them, there is a discrepancy of 20-30% of the results. It is difficult to define which MGA is the best test, and there is also a significant discrepancy between the most widely used ODX and MMP. In this situation, I think the AI model that predicts MGA risk developed by integrating data from patients who underwent the widely used ODX or MMP may be a clue to narrowing the gap between MGA inconsistency results.
Comment: The results report the following: 'In the ensemble cohort (MMP + ODX), the sensitivity, specificity, and precision for predicting the low-risk category were 0.92, 0.66, and 0.92, respectively. And those for predicting the high-risk category was 0.66, 0.92, and 0.65, respectively (Figure 2 & Table 4).’ I understand that the models are binary classificatory for predicting the low or high risk category. So why specify sensitivity, specificity and precision for the low-risk class and then for the high-risk class? Clearly define what the positive class (sensitivity and precision) and the negative class (specificity) are. Please explain in the materials section which is the positive reference class and which is the negative one.
Response: As mentioned in the discussion, reliable identification of patients with low risk tumors even when MGA is not readily available is crucial to minimize unnecessary cytotoxic exposure and to reduce the treatment costs. Based on this, we selected the low risk as a positive class and added the explanation to the method section according to your suggestion.
Reviewer 4 Report
Comments and Suggestions for Authors
1. The Introduction section is very concise. This section should contain the Importance of the problem, existing ways for this problem-solving, the limitations of the existing methods, the authors' propositions regarding the problem's solution and the research's main contributions. Thus, this section should be rewritten.
2. Please add the Literature Survey section. This section should contain a brief analysis of the current research in this subject area with an allocation of the unsolved parts of the general problem at the end of this section.
3. Adding a block chart of the research in the section Material and Methods will be better.
4. The section Conclusions should be extended too by adding a brief description of the obtained results, allocating its performance and further prospects of the authors' research.
Author Response
Comment:
- The Introduction section is very concise. This section should contain the Importance of the problem, existing ways for this problem-solving, the limitations of the existing methods, the authors' propositions regarding the problem's solution and the research's main contributions. Thus, this section should be rewritten.
Response: Thank you for your comment. As you mentioned, we modified the Introduction section.
Importance of the problem: high cost of test
Existing ways for this problem-solving: development of the alternative test
Limitations of the existing methods: insufficiency of accuracy
The authors' propositions regarding the problem's solution and the research's main contributions: development of a more accurate and affordable prediction model
- Please add the Literature Survey section. This section should contain a brief analysis of the current research in this subject area with an allocation of the unsolved parts of the general problem at the end of this section.
Response: The results and characteristics of previous studies related to ODX prediction model were described in the discussion section (Line #282~295).
- Adding a block chart of the research in the section Material and Methods will be better.
Response: In response to the feedback, we added a workflow chart of the research in the Method section.
- The section Conclusions should be extended too by adding a brief description of the obtained results, allocating its performance and further prospects of the authors' research.
Response: Thank you for your comment. As you mentioned, we modified the Conclusion section.
Round 2
Reviewer 4 Report
Comments and Suggestions for Authors
Unfortunately, the manuscript was not major revised according to my remarks.
1. The Literature Survey section should contain the current research in this subject area with the allocation of both their advantages and shortcomings. At the end of this section, it will be better to allocate the unsolved parts of the general problem. This section should be after the Introduction. Of course, in the Discussion section, you compared your results with other existing ones, but it does not replace the Literature survey section.
2. In the Conclusions it will be better to add the further prospects of the authors' research.
Author Response
RESPONSES TO THE EDITORIAL REQUESTS
Comment:
- The Literature Survey section should contain the current research in this subject area with the allocation of both their advantages and shortcomings. At the end of this section, it will be better to allocate the unsolved parts of the general problem. This section should be after the Introduction. Of course, in the Discussion section, you compared your results with other existing ones, but it does not replace the Literature survey section.
Response: Thank you for your valuable comments. As you suggested, we added ‘Related Work’ section after the Introduction.
- In the Conclusions it will be better to add the further prospects of the authors' research.
Response: We added the further prospects of our study in Conclusion. Based on this study, we are preparing a study for a more feasible model. Thank you.